# The Role of Ergosterol and Sphingolipids in the Localization and Activity of *Candida albicans*’ Multidrug Transporter Cdr1p and Plasma Membrane ATPase Pma1p

**DOI:** 10.3390/ijms23179975

**Published:** 2022-09-01

**Authors:** Aneta K. Urbanek, Jakub Muraszko, Daria Derkacz, Marcin Łukaszewicz, Przemysław Bernat, Anna Krasowska

**Affiliations:** 1Faculty of Biotechnology, University of Wroclaw, 50-383 Wroclaw, Poland; 2Department of Industrial Microbiology and Biotechnology, Faculty of Biology and Environmental Protection, University of Lodz, Banacha 12/16, 90-237 Lodz, Poland

**Keywords:** *Candida albicans*, ergosterol, sphingolipids, Cdr1, H^+^-ATPase

## Abstract

Opportunistic pathogen *Candida albicans* causes systemic infections named candidiasis. Due to the increasing number of multi-drug resistant clinical isolates of *Candida* sp., currently employed antifungals (e.g., azoles) are insufficient for combating fungal infection. One of the resistance mechanisms toward azoles is increased expression of plasma membrane (PM) transporters (e.g., Cdr1p), and such an effect was observed in *C. albicans* clinical isolates. At the same time, it has been proven that a decrease in PMs sphingolipids (SLs) content correlates with altered sensitivity to azoles and diminished Cdr1p levels. This indicates an important role for SL in maintaining the properties of PM and gaining resistance to antifungal agents. Here, we prove using a novel spot variation fluorescence correlation spectroscopy (svFCS) technique that CaCdr1p localizes in detergent resistant microdomains (DRMs). Immunoblot analysis confirmed the localization of CaCdr1p in DRMs fraction in both the *C. albicans* WT and *erg11Δ/Δ* strains after 14 and 24 h of culture. We also show that the *C. albicans*
*erg11Δ/Δ* strain is more sensitive to the inhibitor of SLs synthesis; aureobasidin A (AbA). AbA treatment leads to a diminished amount of SLs in *C. albicans* WT and *erg11Δ/Δ* PM, while, for *C. albicans*
*erg11Δ/Δ*, the general levels of mannose-inositol-P-ceramide and inositol-P-ceramide are significantly lower than for the *C. albicans* WT strain. Simultaneously, the level of ergosterol in the *C. albicans* WT strain after adding of AbA remains unchanged, compared to the control conditions. Analysis of PM permeabilization revealed that treatment with AbA correlates with the disruption of PM integrity in *C. albicans*
*erg11Δ/Δ* but not in the *C. albicans* WT strain. Additionally, in the *C. albicans* WT strain, we observed lower activity of H^+^-ATPase, correlated with the delocalization of both CaCdr1p and CaPma1p.

## 1. Introduction

*Candida albicans* constitutes a part of the normal microbiota of healthy individuals [1]. In case of immunodeficiency (e.g., HIV infection) *C. albicans* can lead to life-threatening systemic infections (candidiasis). A limited number of clinically employed antifungal therapeutics translates into increasing resistance of *Candida* sp. [2]. Azoles (e.g., fluconazole and miconazole), one of the most commonly used antifungals, targets the enzyme Erg11p (encoded by the *ERG11* gene), which is responsible for the removal of the C-14α-methyl group from lanosterol in ergosterol biosynthesis pathway [3]. The disruption of biosynthesis corresponds to the diminished plasma membrane (PM) ergosterol content and pathogen growth inhibition. On the other hand, decades of azole usage led to the formation of multiple resistance strategies of *C. albicans*. These include the upregulated expression of *ERG11* (via point mutation in *UPC2*, a transcription factor for *ERG11* gene, or in *ERG11* directly) and genes encoding *C. albicans* drug resistance transporters (e.g., *CDR1*, *CDR2*, and *MDR1*) [4]. Efflux pumps, Candida drug resistance protein 1 and 2 (CaCdr1p and CaCdr2p, respectively), belong to the ABC transporters family and azoles are substrates for these proteins. Furthermore, the overexpression of *CDR1* and *CDR2* was detected in azole-resistant oral and vaginal *C. albicans* clinical isolates [2,5].

Still, not all of the drug resistance strategies of *C. albicans* are fully understood. Interestingly, it has been proven that the inhibition of the mannose-(inositol phosphate)2-ceramide (M(IP)_2_C) biosynthesis, a sphingolipid (SL) present in *C. albicans* PM, correlates with significantly elevated susceptibility to azoles [6]. In addition, the deletion of the *IPT1* gene (encoding inositol phosphotransferase required for biosynthesis of M(IP)_2_C) in *C. albicans* resulted in a decreased level of Cdr1p in fungal PM [7]. This suggests that SLs play a significant role in maintaining the PM structure and proper functioning of multidrug resistance transporters. Thus, it is relevant to further investigate the importance of SLs in CaCdr1p activity. SLs participate in the creation of ordered domains, as well as the segregation of fungal PM components [8]; however, SL-enriched domains (SLEDs) were extensively studied only in *Saccharomyces cerevisiae* model organisms [9].

Here, we demonstrate that Cdr1p localizes in the lipid rafts of both the *C. albicans* WT and ergosterol deficient (*erg11Δ/Δ*) strains using a novel spot variation fluorescence correlation spectroscopy (svFCS) technique [10], which was confirmed by the analysis using Optiprep^tm^ and Western Blot. CaCdr1p was found in the same fractions as plasma membrane H^+^-ATPase (CaPma1p-GFP), which naturally concentrate in the detergent-resistant microdomains (DRMs) of *S. cerevisiae* and have been used as a marker for DRMs in other fungi [11,12]. Following the treatment of *C. albicans* strains by aureobasidin A (AbA), a cyclic depsipeptide that inhibits the activity of the catalytic subunit of IPC synthase (Aur1) [13], we observed a significant decrease in both the IPC and MIPC levels in PM.

Disruption of the PM structure (lipid rafts) resulted in the delocalization of CaCdr1p and CaPma1p from PM to the inside of cells, permeabilization of PM, and a decrease in the activity of the studied proteins.

To our knowledge, here, for the first time in the literature, we show results indicating that both ergosterol and SLs are crucial for the maintenance of *C. albicans*’ PM structure. Moreover, ergosterol partially compensate for the lack of SLs in *C. albicans*’ PM.

The obtained data open new paths for further research on the role of SLs in *Candida* sp. resistance to antifungals involving multidrug transporters, which will provide a better understanding of those mechanisms and contribute to the development of new antifungal therapies.

## 2. Results

### 2.1. CaCdr1p Is Located in Nanodomains of PM of C. albicans

Some disruptive (by use of detergents, sequesters, or genetic approaches) and non-disruptive (e.g., fluorescence correlation spectroscopy (FCS) or fluorescence resonance energy transfer (FRET)) techniques are already used for the structure and function of lipid domains study. It has recently been proven that the svFCS can be used for the detailed examination of the molecular dynamics of plasma membrane lateral organization in living cells [10].

The offset t0 depends on membrane properties, such as lipid composition and actin cytoskeleton organization, which allows for a biological insight into the origin of the observed phenomenon [10,14,15].

The svFCS method enables space–time analysis of labeled single molecules in the cell membrane. The law of diffusion established by the svFCS allows for determining the diffusion times of individual membrane components and the nature of their movement in the plasma membrane. The diffusion time depends on the size of the molecule. The larger the molecule, the longer the diffusion time. The nature of diffusion depends on intermolecular interactions. In the absence of interactions, the free diffusion characteristic of the Brownian motion is observed. In such diffusion, the point of intersection with the 0Y (t0 value) is close to 0. In the case of interactions, for example, the molecule’s location in the nanodomain is greater than zero [16].

Cdr1p-GFP in both C. albicans strains (CAF2-1—70 ms and erg11∆/∆—65 ms) has a positive t0 value (Figure 1A,B), indicating the localization of this transporter in the PM nanodomains. The high t0 value also reflects the relatively high labeled Cdr1p-GFP protein size. In the case of the C. albicans erg11∆/∆ strain, the t0 value is about 5 ms lower than in the CAF2-1 strain. This slight decrease may suggest that Cdr1p-GFP membrane diffusion was changed into a more Brownian, free diffusion mode. This result indicates that the lack of ergosterol does not destroy nanodomains. We could hypothesize that lack of the ergosterol could be somehow compensated in the nanodomains by SLs, resulting in only a small difference in the diffusion properties of Cdr1p-GFP.

Although the svFCS method seems suitable for analyzing the localization of CaCdr1p in lipid nanodomains, it is good to use an alternative method to make the results more plausible. For this purpose, we obtained fractions of the PM, and then we visualized the localization of CaCdr1p and CaPma1p in individual fractions by Western Blot using rabbit anti-GFP serum (Figure 2).

As described previously [17,18], DRM usually constitutes the upper two fractions when the equal six fractions from top to bottom are taken. Pasrija et al. [18] used 30% Optiprep^tm^ in their study. Pma1p was previously localized in the PM microdomains of *S. cerevisiae* by Bagnat et al. [12]; hence, it can be considered an indicator of the localization of other membrane proteins, e.g., Cdr1p.

Our immunoblot analysis indicated the presence of both CaPma1p-GFP and CaCdr1p-GFP, mainly in two floating top fractions of 30% Optiprep^tm^ (Figure 2B) in the case CAF2-1 strain in 8, 14, and 24 h of culture. We used the division into three fractions in the 30% gradient of Optiprep^tm^ and observed the presence of Pma1p in the third fraction in all culture times. Cdr1p appeared in the third fraction only at 24h of the *C. albicans* culture (Figure 2B, left panel). In the case of a mutant without ergosterol (*erg11∆/∆*), Pma1p-GFP was present in three fractions of 30% Optiprep^tm^. However, in the early phase of growth (8 h), *C. albicans* Pma1p occurs in DRM at a low level (Figure 2B, right panel).

We did not detect the presence of Cdr1p in DRM in the early growth phase (8 h) of *erg11∆/∆ C. albicans*. Cdr1p was localized in DRM at 14 h growth of this strain. In the late (24 h) phase of the growth of the *erg11∆/∆* strain, we observed a much smaller level of Cdr1p than in 14 h growth, as well as compared to the level of Pma1p after 24 h of mutant culture without ergosterol (Figure 2B, right panel). The results shown in the immunoblot correlate with the results obtained with svFCS. In both methods, it was shown that, in the *erg11**Δ/Δ* mutant, after 24 h of culture, there is less Cdr1p in the nanodomains/DRM than in the WT strain. In addition, in WT, the Cdr1 protein is present in a larger amount of fraction than in the *erg11**Δ/Δ* strain, which may indicate that, as a result of the changes in membrane diffusion, part of the protein has been delocalized from the membrane.

The preliminary test, with an example of the protein content in the fractions collected from the DRM isolation procedure with Optiprep^tm^ gradient (*erg11∆/∆* strain, 24 h of culturing), indicated that the most of the Pma1p localizes in the fraction containing 30% of Optiprep^tm^ (Appendix A). Ponceau S staining reveals accumulation of the most of isolated proteins onto top on the gradient, which, along with similar localization of Pma1p, confirms efficient isolation of proteins associated with DRMs (Appendix A). Cdr1p localizes together with Pma1p, suggesting its accumulation in DRMs (Appendix A).

### 2.2. Low Sphingolipid Level in the PM Strongly Inhibits the Growth of C. albicans Mutant without Ergosterol (erg11∆/∆)

Bagnat et al. [12] found that both sphingolipids and ergosterol, which are constituents of DRMs, may be necessary for the proper localization of the Cdr1 protein in PM. They also observed that the deletion of genes encoding enzymes for ergosterol biosynthesis and sphingolipid pathways leads to the mislocation of Cdr1p, as well as its decreased activity. The above study was conducted on non-pathogenic yeast *S. cerevisiae* containing genes from *C. albicans*. To the best of our knowledge, there is a lack of information on the effect of the lipid composition in DRMs on Cdr1p directly in pathogenic *C. albicans* fungi, so we decided to conduct appropriate research.

Previously, our results showed frail growth of the mutant *C. albicans* without ergosterol (*erg11∆/∆*), in comparison to the WT strain (CAF2-1), when grown in complex YPD under aerobic conditions [19]. Mukhopadhyay et al. [7] found that the *C. albicans* mutants, with deletions of the *ERG6* and *ERG16* genes of the ergosterol biosynthesis pathway, are supersensitive to many drugs, including 4-nitroquinoline, oxide, terbinafine, and o-phenanthroline, as well as to itraconazole and ketoconazole. Our results indicated that the deletion of the *ERG11* gene encoding the lanosterol demethylase, which is a target for azoles, causes a loss of sensitivity of *C. albicans* to azoles [19] and hypersensitivity to compounds such as capric acid [20] or surfactin [21].

Results presented in Figure 3 indicate high lethality of the ergosterol-free *C. albicans* strain (*erg11∆/∆*), under the influence of aureobasidin A (AbA). AbA is an inhibitor of the synthesis of inositol-P-ceramide (IPC), sphingolipid, which is a constituent of PM and precursor of other PM sphingolipids. At a concentration of 0.0031 μg/mL of AbA, the *erg11∆/∆* strain is almost one and a half orders more sensitive than the CAF2-1 strain (Figure 3B). Minimal inhibitory concentrations of 50 and 90 (MIC50 and 90) for mutant *erg11∆/∆* were estimated at 0.007 and 0.01 μg/mL and 7- and 10-fold lower than for WT strain, respectively (Figure 3A).

### 2.3. Aureobasidin a Changes the Lipid Composition in PM of C. albicans

The loss of ergosterol results in the destabilization of PM, its decreased permeability, reduced fluidity, and decreased potential (∆ψ) [19]. In addition to ergosterol, the second component of *C. albicans* DRM are sphingolipids (SLs), which structurally differ from those in mammalian cells [7]. Hypothetically SLs structures are synthesized by cells to compensate for the loss of ergosterol in the membrane by matching in terms of polarity. In *S. cerevisiae*, it was observed that mutants deprived of the genes involved in ergosterol synthesis increased SLs synthesis with smaller polar head groups (e.g., mannose-inositol-P-ceramide and inositol-P-ceramide (MIPC + IPC)) and decreased SLs synthesis with the largest polar head group (mannose-(inositol-P)_2_-ceramide (M(IP)_2_C)) [22].

Therefore, we were interested in investigating the individual SL classes in *C. albicans*. We checked the presence and concentration of IPC, MIPC, and M(IP)_2_C in the *C. albicans* CAF2-1 (WT strain) and mutant without ergosterol KS028 (*erg11∆/∆*) during treatment with and without AbA.

In the CAF2-1, the ergosterol level increases in the logarithmic phase of growth and decreases in the stationary phase. These data are consistent with our earlier results [19], and a similar effect was observed in cases of other fungi [23]. Thus, it can be assumed that such an effect is universal. In the CAF2-1 strain, the ergosterol level slightly decreased after AbA treatment (Figure 4A). The concentration of both IPC and MIPC decreased with the ageing of the CAF2-1 culture. In all tested phases of the growth of *C. albicans* CAF2-1, the level of IPC and MIPC significantly decreased (Figure 4B). On the other hand, in the KS028 strain, the decrease in the level of IPC during the ageing of the culture was not observed. However, the amount of MIPC was dropped (Figure 4B). The addition of AbA strongly reduced the level of IPC and MIPC in all phases of KS028 growth (Figure 4B). After 8 and 14 h of cultivation, KS028 had about 50% less IPC in PM than CAF2-1. In the stationary growth phase, the KS028 strain had about 15% less IPC than the WT strain. The differences in the MIPC levels in both studied strains were much smaller than in IPC (Figure 4B). We did not find M(IP)_2_C in the tested samples.

Analysis of the molecular species of sphingolipids revealed nine species of IPC and MIPC in samples (Appendix A). These species are represented as the “total number of carbons in the sphingoid backbone and the fatty acyls:total number of double bonds;total number of hydroxyl groups in the sphingoid backbone and the fatty acyls” [24]. Among them, in all samples, IPC 42:0;4 and MIPC 42:0;4 dominated. There was a shift in the molecular species of both IPC and MIPC in *C. albicans* CAF2-1 older cultures. At 24 h, compared to 8 h, the ratio of longer-chain (such as 44:0;4) to shorter chain species (such as 42:0;4) was higher. Such phenomenon was not so clearly visible in KS028 (*erg11Δ/Δ*).

### 2.4. The Lack of Ergosterol but Not the Reduced Level of Sphingolipids in C. albicans’ PM Caused the Delocalization of CaCdr1p and CaPma1p to the Inside of the Cells at Different Times

In our previous studies, we observed the delocalization of CaPma1p and CaCdr1p, from PM to vacuole, after 8 h of the culture of the strain without ergosterol, unlike the WT strain, in which such delocalization took place only after 14 h of culture [19].

In this work, we present results from the localization of Pma1p-GFP and Cdr1p-GFP in strains *C. albicans* WT (YHXW11 and AsCa1) and *erg11Δ/Δ* (KS045 and KS023), as well as in the presence of AbA, with a reduced level of SLs (IPC and MIPC) in PM. We also compare the presence of GFP-labelled Pma1 and Cdr1 proteins in PM in confocal microscope’s images and Western Blot gels (Figure 2 and Figure 5).

As in the previous results, we observed the delocalization of both CaPma1p and CaCdr1p from PM to the inside of cells, with temporal differences between the culture phases of the WT strain and strain without ergosterol. It seems that the addition of 0.005 μg/mL AbA did not accelerate the delocalization of CaPma1p and CaCdr1p from PM. Such a shift of the studied proteins was visible in the WT strains only in 14 h of culture. In the case of strains without ergosterol, no increased level of proteins was observed inside the cells in 8 h of culture (Figure 5).

Interestingly, both CaPma1p and CaCdr1p in the photos from 8 h of culturing were clearly found in PM both in WT and in the strain without ergosterol, while, on the Western Blot, we did not observe the presence of these proteins in 8 h culture in the DRM strain *erg11Δ/Δ* (Figure 5 control 8 h, Figure 2B right panel).

In 24 h of culture, we observed a reduced level of CaPma1p in the PM strains WT and *erg11Δ/Δ,* as well as the lack of CaCdr1p in the PM of these strains (Figure 5, control 24 h). This result was supported by the data obtained from Western Blot, which confirmed the presence of CaPma1p and CaCdr1p after 24 h of culture. However, the intensity was much lower than that obtained after 14 h of culture (Figure 2). At the same time, in strain *erg11Δ/Δ*, the level of Cdr1 protein seemed much less than Pma1p on the Western Blot (Figure 2, panel B).

### 2.5. Lack of Ergosterol as Well as Reduced Sphingolipid Level in PM Reduce Cdr1p and H^+^-ATPase Activity

We have previously shown that Cdr1p activity in a mutant without ergosterol (KS028) is lower than in the WT strain (CAF2-1) [19]. Here, we investigated how the functioning of CaCdr1p will be affected by the reduced level of sphingolipids (after AbA treatment) in PM. Firstly, we investigated whether AbA usage contributes to the permeabilization of *C. albicans* CAF2-1 or KS058 by propidium iodide (PI) staining (Figure 6). Simultaneously, an R6G efflux assay was used to determine the Cdr1p activity (Figure 7). In the case of the CAF2-1 strain, the addition of AbA caused an increase in fluorescence after 14 and 24 h incubation, which may suggest an increase in Cdr1p activity (Figure 7). We observed progressive permeabilization of PM in the CAF2-1 strain during culturing (Figure 6), especially after 24 h, which may be the cause of rhodamine outflow, not only by Cdr1p, but also partly by PM, which gave increased fluorescence in the tested samples.

After the addition of AbA, we observed a very high permeabilization of PM in strain KS028, as measured by a PI assay (Figure 6). In turn, we observed a very low fluorescence of rhodamine, which may suggest low Cdr1p activity or the lack thereof (Figure 7). The low activity of Cdr1p in the KS028 strain under the influence of AbA can be confirmed by the observed delocalization of this transporter from PM to the inside of the cell under these conditions (Figure 5).

In a previous study in the *C. albicans* strain without ergosterol, we observed both lower Cdr1p and H^+^-ATPase activity, compared to the CAF2-1 strain [19]. In this work, we investigated the activity of Pma1p after the addition of AbA using the measurement of acidification outside the cells of *C. albicans*. In almost all samples, we observed a decrease in the intensity of acidification of the environment by cells under the influence of AbA. Only the CAF2-1 strain in the early growth phase (8 h) acidified the environment more intensively in the presence of AbA than without this compound (Table 1). The inhibition of the rate of acidification of the environment by Pma1p by AbA indicates a decrease in the activity of this protein. The results shown in Figure 5 suggest the additional synthesis of Pma1p, especially by the strain without ergosterol. Despite the delocalization of Pma1 from PM to the inside of cells during the cultivation in the presence of AbA, H^+^-ATPase is also found in PM. It seems that, after 24 h of culture and in the presence of AbA, in the PM strain without ergosterol, the level of Pma1 protein is higher than in the parental strain (Figure 5). Hence, the activity of this protein is less inhibited in KS028 than in the CAF2-1 strain (Table 1).

## 3. Discussion

Localization of membrane proteins in the plasma membrane (PM) microdomains is a suggested mechanism to explain the effect of membrane lipids on fungal virulence [25]. Although lipid nanodomains are still a controversial term, it is increasingly accepted that domains composed of sphingolipids and sterols exist in cell membranes. Much information about lipid microdomains has been obtained in model membrane studies, and only very few studies have focused on lipid microdomains in pathogenic fungi [26]. Kohli et al. [27] used non-pathogenic yeast *S. cerevisiae*, with an overexpressed CaCdr1p transporter. They found that reduced levels of ergosterol in the membrane cause an increased expression of the *CDR1* gene and increased resistance to fluconazole. Pasrija et al. [18] observed that defects in genes encoding the enzymes involved in ergosterol or sphingolipids biosynthesis cause the defective transport of Cdr1p to the *S. cerevisiae* plasma membrane. According to the above, the non-pathogenic yeast *S. cerevisiae* seems to be a useful model strain for research. However, according to our results so far, often the processes that take place directly in the pathogenic strain of *C. albicans* differ from those in *S. cerevisiae* with the overexpression of pathogen genes, and this is due to differences in metabolism between these microorganisms [28]. We have shown that the CaCdr1 transporter is located in the DRMs of *C. albicans*. We used the method with cell membrane fractionation in the Optiprep^tm^ gradient and immunoblot analysis. We also developed a new method svFCS to study the localization of CaCdr1p. Using this technique, we proved the living *C. albicans* cells presence of CaCdr1p in PM microdomains (Figure 1). Thus, we confirmed the results obtained by Pasrija et al. [18] that used *S. cerevisiae* with CaCdr1p overexpression.

The microdomains in the PM consist of both ergosterol and sphingolipids [12]. If the PM of *C. albicans* is deprived of ergosterol and sphingolipid levels are reduced, it seems that microdomains will not be created or do not perform their functions properly. In this work, we wanted to observe whether the Cdr1 transporter, as well as H^+^-ATPase, would be invariably located in the PM, despite the absence of microdomains. The microscopic study of *C. albicans* Cdr1p-GFP and Pma1p-GFP revealed the delocalization of both proteins from PM to the inside of the cells. In the mutant without ergosterol, proteins delocalization is visible already after 8 h of culture growth, in contrast to WT strains, in which this process was observed only after 14 h (Figure 5). Unexpectedly, despite the lack of ergosterol and reduced levels of sphingolipids, Cdr1p and Pma1p maintained the localization in PM in 14 h of culture. Moreover, it seems that in 24 h of culture, a certain amount of Pma1p was still present in PM of *C. albicans erg11Δ/Δ* mutant, even after treatment with AbA (Figure 5). The obtained results suggest that the existence and proper structure of the microdomains are not necessary for maintaining the location of CaCdr1p and CaPma1p in the PM.

Another question was whether CaCdr1p and CaPma1p are located in the PM, despite the lack of ergosterol and reduced sphingolipid levels that are still active. The obtained results showed a decrease in H^+^-ATPase activity under the influence of AbA, but even in the mutant without ergosterol after 24h of culture with AbA, low activity of this protein was observed (Table 1). It seems that CaCdr1p in the mutant without ergosterol displayed much lower activity than in the WT strain, and the value decreased to zero after 24 h of culture. In cells without ergosterol and with low sphingolipid content, we observed very low or lack of activity of CaCdr1p in the early logarithmic phase of growth (8 h) (Figure 7). In the past, another research team has attempted to answer the question of whether the loss of Cdr1 protein localization in the membrane domain causes a loss of its function and their results seemed to confirm this [28]. The results obtained by our team indicate that the lack of ergosterol and reduced level of SLs allow for the partial persistence of Cdr1p and Pma1p in PM, but reduced the activity of these proteins. It can be concluded that improperly constructed membrane domains interfere with such processes as, for example, the fluidity of membranes. That, in turn, disrupts the molecular interactions of proteins and lipids necessary for PM functioning and activity. Despite observations so far, we are still far away from being able to complete the cognition and description of the molecular environment of proteins in the lipid domains.

## 4. Materials and Methods

### 4.1. Chemicals

Chemicals used in this study were obtained from the following sources: peptone, yeast extract (manufacturer: BD; distributor: Life Technologies; Warszawa, Poland), aureobasidin A (AbA) (Takara Bio USA; San Jose, CA, USA), bacteriological agar, zymolyase, D-glucose, D-sorbitol (manufacturer: BioShop; distributor: Epro Science, Puck, Poland), ethylenediaminetetraacetic acid (EDTA), phosphate-buffered saline (PBS), β-mercaptoethanol (BME), 2-deoxy-D-glucose, rhodamine 6G (R6G) (Merck Life Science; Poznań, Poland), 08:0 PI (1,2-dioctanoyl-sn-glycero-3-phospho-1′-myo-inositol, ammonium salt) (Avanti phospholipids, Birmingham, AL, USA), ergosterol, lanosterol, cholesterol, hexane, deionized water and BSTFA/TMCS (N,O-bis(trimethylsilyl) trifluoroacetamide/trimethylchlorosilane) (Merck, Germany) chloroform (CHCl_3_), methanol (MeOH), KOH, and HCl (Chempur; Piekary Śląskie, Poland).

### 4.2. Strains and Growth Conditions

Strains of *C. albicans* that were used in this study are listed in Table 2.

**Table 2 ijms-23-09975-t002:** *C. albicans* strains used in this study.

Strain	Genotype	Reference
CAF2-1	*ura3∆::imm434/URA3*	[29]
KS023	*ura3∆::imm434/ura3∆::imm434 CDR1/CDR1-yEGFP-URA3 erg11∆::SAT1-FLIP/erg11∆::FRT*	[19]
KS028	*ura3∆::imm434/URA3 erg11∆::SAT1-FLIP/erg11∆::FRT ura3∆::imm434/ura3∆::imm434*	[19]
AsCa1	*ura3∆::imm434/ura3∆::imm434 CDR1/CDR1-yEGFP-URA3*	[30]
YHXW11	*ura3∆::imm434/ura3∆::imm434 PMA1/PMA1-GFPγ-URA3*	[31]

In all cases, *C. albicans* strains were grown in YPD agar plates (1% peptone, 1% yeast extract, 2% glucose, and 2% agar) and pre-grown in YPD medium (24 h, 28 °C, 120 rpm). For experiments, 20 mL of YPD medium (with or without AbA) was inoculated with pre-culture, starting at A_600_ = 0.1 and grown for 8, 14, or 24 h (28 °C, 120 rpm).

### 4.3. Determination of Minimal Inhibitory Concentration (MIC) and Spot Tests

In order to determine the MIC50 and MIC90 values, a serial dilution of AbA (0–1.6 μg/mL) was prepared in YPD medium using sterile 96-well plates (Sarstedt, Nümbrecht, Germany). Then, wells containing YPD medium were supplemented with AbA and then inoculated with *C. albicans* CAF2-1 and KS028 strains (final A_600_ = 0.01). Plates were incubated overnight at 28 °C. After this, optical density (λ = 600 nm) was measured using Asys UVM 340 (Biogenet). Negative controls were wells with YPD medium alone. The MIC50 and MIC90 values state the lowest concentration of AbA that led to 50% and 90% decreases in *C. albicans* growth. The experiment was performed in 3 technical (within the same assay) and biological replicates (*n* = 3).

Spot tests were performed using solid YPD medium containing (or not, in case of control) selected concentrations of AbA. Then, overnight, pre-culture of *C. albicans* CAF2-1 and KS028 strains were spotted (3 μL) on YPD agar plates in serial dilution, starting from A_600_ = 0.7 for both strains. Plates were grown for 48 h at 28 °C, and photographs were taken using FastGene B/G GelPic LED Box (Nippon Genetics Europe GmbH, Düren, Germany).

### 4.4. Isolation of Plasma Membrane

PMs were obtained as previously described [32]. Briefly, *C. albicans* CAF2-1 and KS028 cells were harvested (3 min, 6000 rpm) after 8, 14, or 24 h of culture (equal A_600_ = 40 for all used strains) and lysis solution was added to the cells (1 M sorbitol, 0.1 M EDTA, 1% BME, 3 mg/mL zymolyase). After 30 min of incubation at 37 °C, cold H_2_0_dd_ was added, and protoplasts were sonicated (5 s cycles, 2 min each) using an ultrasonic processor (Heilser UP50H). Next, lysates were centrifuged (10 min, 10,000 rpm, 4 °C), and the supernatant was then ultracentrifuged for 1 h, at 100,000 rpm, at 4 °C using micro ultracentrifuge CS150FNX (Hitachi; Tokyo, Japan). Pellets of PMs were resuspended in PBS, and mixture of CHCl_3_–MeOH (1:2 *v/v*) was added. Finally, after continuous shaking for 24 h (4 °C), a phase containing CHCl_3_ was added to glass vials and concentrated with nitrogen gas. The PM isolation was performed in 3 independent, biological replicates (*n* = 3).

### 4.5. Sphingolipids Determination

Sphingolipid species were detected in base hydrolyzed lipid samples (Section 4.4) using LC-MS/MS system (*n* = 3). Extracted lipids were suspended in 500 μL methanol. The UHPLC system (ExionLC Ac, Sciex) was connected to a hybrid triple quadrupole linear ion trap mass spectrometer QTRAP 4500 (Sciex) equipped with a Turbo V source ion spray operating in negative ESI mode. Gradient chromatographic separation was performed on an ZORBAX Eclipse XDB (Agilent, USA) C18 column (50 × 2.1 mm), with a 1.8 μm particle size. The injection volume was 10 μL, and the column was maintained at 37 °C. The mobile phase consisted of water containing 5 mM ammonium formate (eluent A) and methanol containing 5 mM ammonium formate (eluent B), with the flow rate of 500 μL/min. Gradient elution was performed with 70% B for 0.25 min, a linear increase to 100% B until 1 min, 100% B until 7 min, and reequilibration from 7 to 7.1 min, with 70% B, and held to 9 min. The turbo ion spray source was operated in the negative ionization mode using the following settings: ion spray voltage = −4500 V, ion source heater temperature = 600 °C, source gas 1 = 55 psi, source gas 2 = 45 psi, and curtain gas setting = 25 psi. To perform the sphingolipids species survey, an information-dependent acquisition method was prepared. Lipid species were analyzed using negative precursor ion scanning, using *m*/*z* 241 and *m*/*z* 259 for IPC and *m*/*z* 421 for MIPC, respectively, as the precursor ion, and the spectra were obtained over a range from *m*/*z* 100 to 1500. Based on the product ion and precursor ion analysis a list of MRM transitions was then generated. The sphingolipids were quantified, in comparison to 08:0 PI (1,2-dioctanoyl-sn-glycero-3-phospho-(1′-myo-inositol) (ammonium salt), Avanti phospholipids, USA).

### 4.6. GC-MS Sterols Analysis in Plasma Membrane

The work-up procedure was previously described [33]. We added to the concentrated lipid extracts (Section 4.4.; *n* = 3) 0.5 mL CHCl_3_, 0.5 mL MeOH-KOH (0.6 M), and 20 μL cholesterol solution in CHCl_3_ (calibration standard, 1 mg/mL). Samples were vortexed and incubated at 23 °C for 1 h. Then, 0.325 mL 1M HCl and 0.125 mL H_2_O were introduced and centrifuged (5000 rcf; 10 °C; 5 min). The lower chloroform layer containing the lipids was transferred to 1.5 mL Eppendorf tubes and dried. Then, 100 μL of silylation reagent BSTFA+TMCS was added and heated for complete silylation at 85 °C for 90 min. The cooled samples had 50 µL hexane introduced and vortexed. Analysis was performed with a gas chromatograph (Agilent 7890) equipped with column HP 5 MS (30 m × 0.25 mm inner diameter, i.d. × 0.25 mm film thickness, f.t.) and 5975C mass detector. The column was maintained at 100 °C for 0.5 min^−1^, then increased to 240 °C at a rate of 25 °C min^−1^, and finally to 300 °C at a rate of 3 °C min^−1^ (for 5 min), with helium as a carrier gas at a flow rate of 1 mL·min^−1^ [33]. The injection port temperature was 250 °C. Sterols were analyzed as trimethylsilyl (TMS) ethers. Ergosterol and lanosterol were analyzed with reference to retention times and fragmentation spectra for standards. Other sterol TMS ethers were identified by comparison with the NIST database or literature data and quantitated using a standard curve for lanosterol.

### 4.7. Determination of Diffusion Time for Cdr1-GFP

For the analysis of diffusion times of Cdr1-GFP protein in *C. albicans* AsCa1 and KS023, strains were cultured in YPD medium for 24 h (28 °C, 120 rpm). Then, cells were harvested (5 min, 4500 rpm), washed with PBS and subjected to LabTEKtm slides (Thermo Fisher Scientific; Warszawa, Poland). Samples were analysed using spot variation fluorescence correlation spectroscopy (svFCS) microscopy [10]; for each waist, at least 20 cells were measured 20 times. Diffusion time for Cdr1-GFP was calculated using IGOR and DiffusionLawsCIML software.

### 4.8. Isolation of DRMs from C. albicans Cells

DRMs were isolated according to Insenser et al. [34], with modifications. *C. albicans* strains were cultured for 8, 14, and 24 h in YPD medium; then, approximately OD_600_ = 100 of cells were pelleted, washed twice in 0.9% NaCl, and frozen in liquid nitrogen. Cells were resuspended in 500 µL of TNE buffer (50 mM Tris pH 7.5, 150 mM NaCl, 5 mM EDTA), followed by the addition of the same volume of 0.45 mm glass beads. Cells were broken by bead beater, glass beads were removed, and the lysate was cleared by centrifugation (1000 rpm, 3 min, 4 °C). Lysate was transferred to a new tube, with the addition of Triton-X100 to final concentration of 1%, and incubated for 30 min on ice.

A total of 400 µL of sample was pipetted carefully to previously prepared ultracentrifuge tubes with gradient of Optiprep^tm^ (Sigma), (200 µL of 60%, 900 µL of 40%, 900 µL of 30%, 400 µL of 5%, (all wt/vol)). Samples were centrifuged (35,000 rpm, 17 h, 4 °C) in swinging bucked rotor (SW 60Ti). After that, equal fractions of 250 µL were collected from the top of each gradient and denatured with addition of TCA to final concentration of 10%, on ice for 30 min. Samples were centrifuged (12,000 rpm, 15 min, 4 °C), and the pellet was resuspended in 90 µL of 1M Tris-HCl pH = 8.0. Then, 30 µL of 4× UREA buffer (8 M Urea, 20 mM EDTA, 5 mM Tris-HCl pH 7.5, bromophenol blue) was added, and samples were incubated for 30 min at 37 °C.

The samples were run on SDS-PAGE and analysed by Western Blot. Membranes were visualized using mouse anti-GFP antibodies (for Pma1-GFP protein), 1:3000 in 3% milk in PBST buffer and rabbit anti-Cdr1 antibodies, and 1:1000 in 3% milk in PBST buffer. The experiment is a representative of three independent assays, and the presented conditions were resolved in the same gel and cut out into separate lines.

### 4.9. Microscopic Study of Cdr1-GFP Localization

In order to determine the localization of Cdr1-GFP and Pma1-GFP, *C. albicans* strains were grown for 8, 14, or 24 h (28 °C, 120 rpm), with or without supplementation with AbA (0.005 μg/mL). Then, cells were centrifuged (3 min, 6 000 rpm), washed 2 times with PBS, and 4 μL of concentrated cell suspension was added on preparation. Microscopic study was performed on a population of cells (at least 100 cells per each case) using a Leica SP8 LSM microscope (Leica microsystems; Wetzlar, Germany). Excitation for GFP-tagged proteins was performed using argon laser (excitation at λ = 488 nm, emission at the range of λ = 500–540 nm).

### 4.10. Propidium Iodide (PI) Staining of C. albicans Cells

In order to investigate the permeabilization of C. albicans PM after AbA treatment, we performed propidium iodide (PI) staining. C. albicans cells after 8, 14, or 24 h of culture, with or without addition of AbA (0.005 µg/mL), were washed three times with PBS (5 min, 4500 rpm) and stained with PI (0.006 mM) for 5 min, RT. Then, cells were harvested, washed three times with PBS, and concentrated. The samples (n = 100 cells per each case) were visualized using a ZEISS AXIO IMAGER A2.

### 4.11. Cdr1p Efflux Pump Assay

The assay was performed as previously described [35] with modifications. *C. albicans* strains were cultured for 8, 14, or 24 h (YPD, 28 °C, 120 rpm), with or without the addition of AbA (0.005 µg/mL). The cells were harvested by centrifugation (5 min, 4500 rpm), washed twice with sterile distilled water, and once with PBS buffer (pH 7.4). The cells were suspended in fresh PBS buffer to OD_600_ of 1.0 and incubated for 60 min at 28 °C and 200 rpm with 5 mM 2-deoxy-D-glucose. Next, 10 µM R6G was added, and the cell suspension was incubated for 90 min. The cells were harvested by centrifugation (5 min, 4500 rpm) and washed with sterile water and PBS buffer (pH 7.4). The pellet was resuspended in PBS buffer to OD_600_ of 10.0 and pre-incubated for 5 min at 30 °C, 300 rpm. R6G efflux was initiated by the addition of 2 mM glucose. At specified time intervals (0 and 15 min), 400 µL of cell suspension was transferred to a microcentrifuge tube, and the cells were harvested by centrifugation (10,000× *g*, 1 min). A total of 100 µL of supernatant was transferred into microtiter black-bottom 96-well plate (Sarstedt, Nümbrecht, Germany). The R6G fluorescence was measured using Cary Eclipse spectrofluorometer (Agilent Technologies, Santa Clara, CA, USA), with excitation at λ = 529 nm (slit 5) and emission at λ = 553 nm (slit 5). The experiment was performed in three biological replicates for each condition.

### 4.12. Proton Extrusion Assay

The method was based on a *S. cerevisiae* protocol [36], with modifications. The 8, 14, and 24 h cultures of *C. albicans* CAF2-1 and KS028 treated and non-treated with 0.005 µg/mL AbA were centrifuged (4500× *g*; 5 min). Obtained pellets were washed twice with distilled water. Real-time acidification of KS028 and CAF2-1 strains suspensions (H_2_O_dd_; OD_600_ = 3.0; 20 mL) was monitored every 10 s for 12 min using a pH-meter (Eutech Instruments CyberScan PH 5500, ThermoFisher Scientific, Warsaw, Poland), equipped with MiniTrode electrode (manufacturer: Hamilton; distributor: Sigma-Aldrich; Poznan, Poland). In each experiment, pH values at t0 were equal to 7.495 ± 0.4. For clearer presentation, these have been normalized to 7.495. The experiment was performed in 3 biological replicates (*n* = 3).

### 4.13. Statistical Analysis

Statistical analysis of data that were obtain in this study was determined using a Student’s *t*-test (binomial, unpaired) or Tukey HSD post hoc test after the one-way ANOVA (α = 0.05). Data represent the means ± standard errors from at least 3 biological replicates. Microscopic study was performed using 3 biological repetitions (at least 100 cells of each were analyzed) and representative micrographs were presented.

## 5. Conclusions

Based on the performed study, we proved that the Cdr1p of *C. albicans* is localized in the PM microdomains. This conclusion was supported by a novel svFCS technique, which was used for determining the localization of proteins in fungal PM microdomains. The analysis revealed that the absence of ergosterol-sphingolipid microdomains does not exclude the localization of CaCdr1p and CaPma1p in PM. Additionally, the lack of ergosterol and reduced amount of sphingolipids in fungal PM decreases or completely reduces the activity of CaCdr1p and CaPma1p.

## Figures and Tables

**Figure 1 ijms-23-09975-f001:**
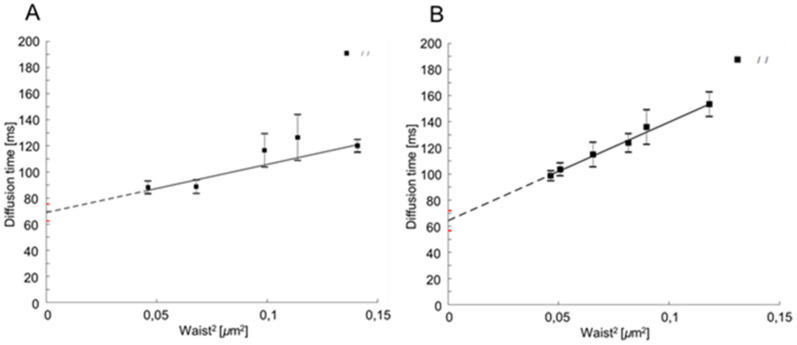
Diffusion time of Cdr1p-GFP, determined in *C. albicans* CAF2-1 (**A**) and *erg11∆/∆* (**B**) strains after 24 h of culture in YPD medium. Diffusion time was measured using svFCS microscopy and then analyzed and calculated with IGOR and DiffusionLawsCIML software.

**Figure 2 ijms-23-09975-f002:**
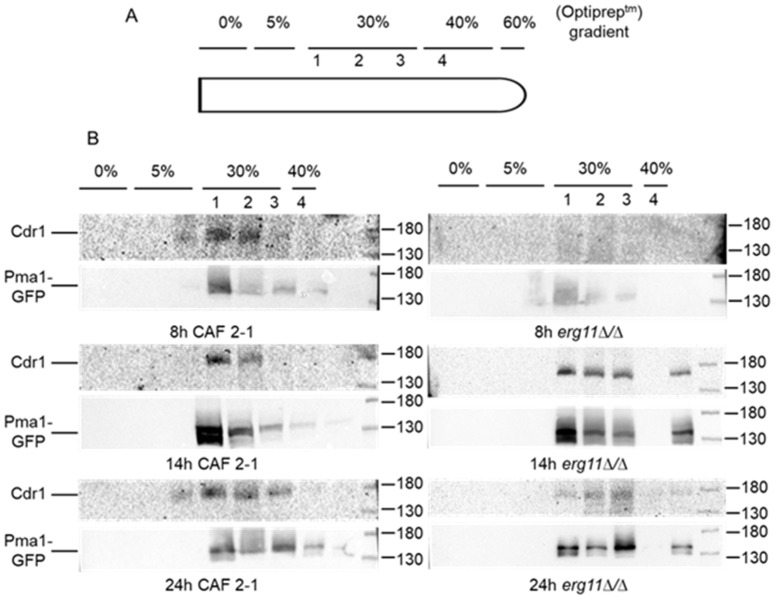
Immunoblot analysis of Cdr1p localization in DRMs fractions from *C. albicans*. (**A**) Schematic representation of used Optiprep^tm^ gradient. (**B**) Wild-type CAF2-1 and ergosterol-deficient *erg11∆/∆* strains were cultured for 8, 14, and 24 h in a YPD medium. Whole-cell extracts were obtained by glass-beads cell disruption, as described in Materials and Methods. Samples were subjected to Optiprep^tm^ gradient (fractions with 0%, 5%, 30% and 40%), followed by ultracentrifugation (35,000 rpm, 17 h). Collected fractions were applied on SDS-PAGE and analyzed by western blot with anti-Cdr1 and anti-GFP antibodies.

**Figure 3 ijms-23-09975-f003:**
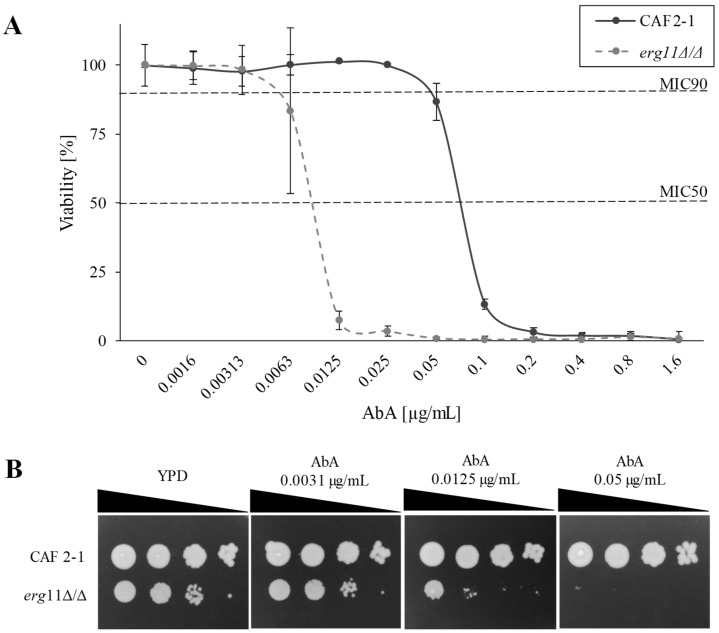
The viability of *C. albicans* CAF2-1 and *erg11∆/∆* strains in the presence of AbA. (**A**) MIC values for CAF2-1 (WT) and KS028 (*erg11∆/∆*) strains in the presence of AbA (0–1.6 μg/mL) after 24 h culture in YPD medium. (**B**) Growth phenotypes of *C. albicans* CAF2-1 and *erg11∆/∆* strains after 48 h of incubation at 28 °C in YPD agar plates, supplemented with AbA.

**Figure 4 ijms-23-09975-f004:**
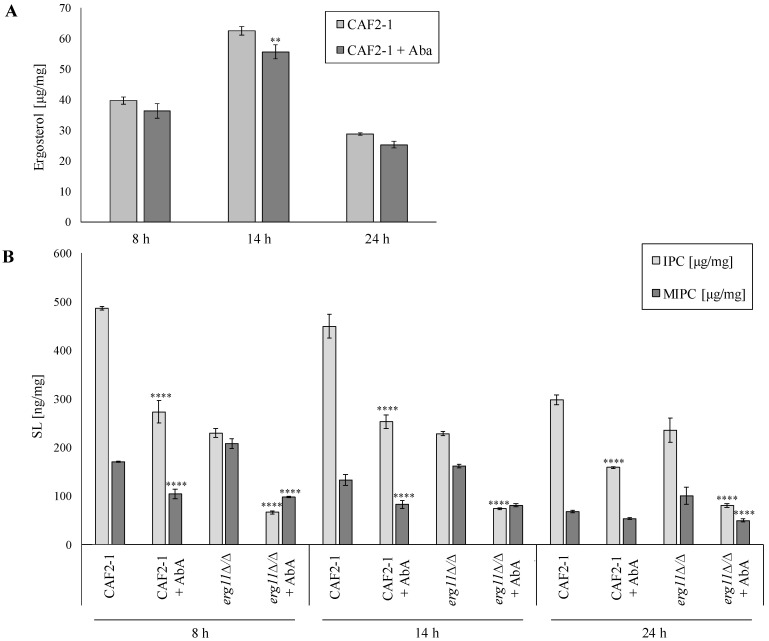
The PMs ergosterol (**A**) and SLs (**B**) analysis (μg/mg and ng/mg (respectively) of the lipids dry mass; ±SD; *n* = 3; ND—not detected) determined for *C. albicans* CAF2-1 and KS028 (*erg11Δ/Δ*) strains in different growth times (8, 14, and 24 h) with AbA (0.005 μg/mL) or without antibiotic (YPD alone). The sterol content was analyzed using the GC-MS method. Statistical analysis was performed by comparing the amounts of specific sterols in *erg11Δ/Δ* against the CAF2-1 strain at each time point and under specific culture condition (** *p* < 0.01, **** *p* < 0.0001).

**Figure 5 ijms-23-09975-f005:**
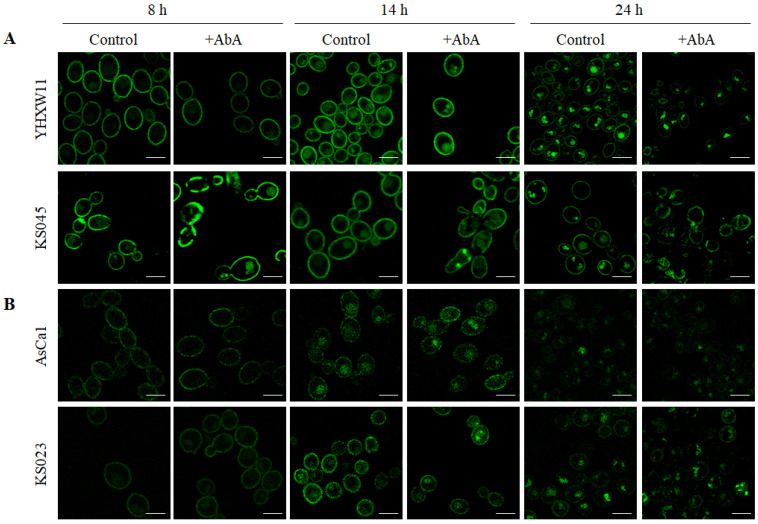
Subcellular localization of Pma1-GFP (**A**) and Cdr1-GFP (**B**) in *C. albicans* CAF2-1 (YHXW11 and AsCa1) and *erg11∆/∆* (KS045 and KS023) strains after 8, 14, and 24 h of culture under control conditions or after treatment with AbA (0.05 μg/mL). Microphotographs were obtain using confocal microscopy. Scale bar 5 μm.

**Figure 6 ijms-23-09975-f006:**
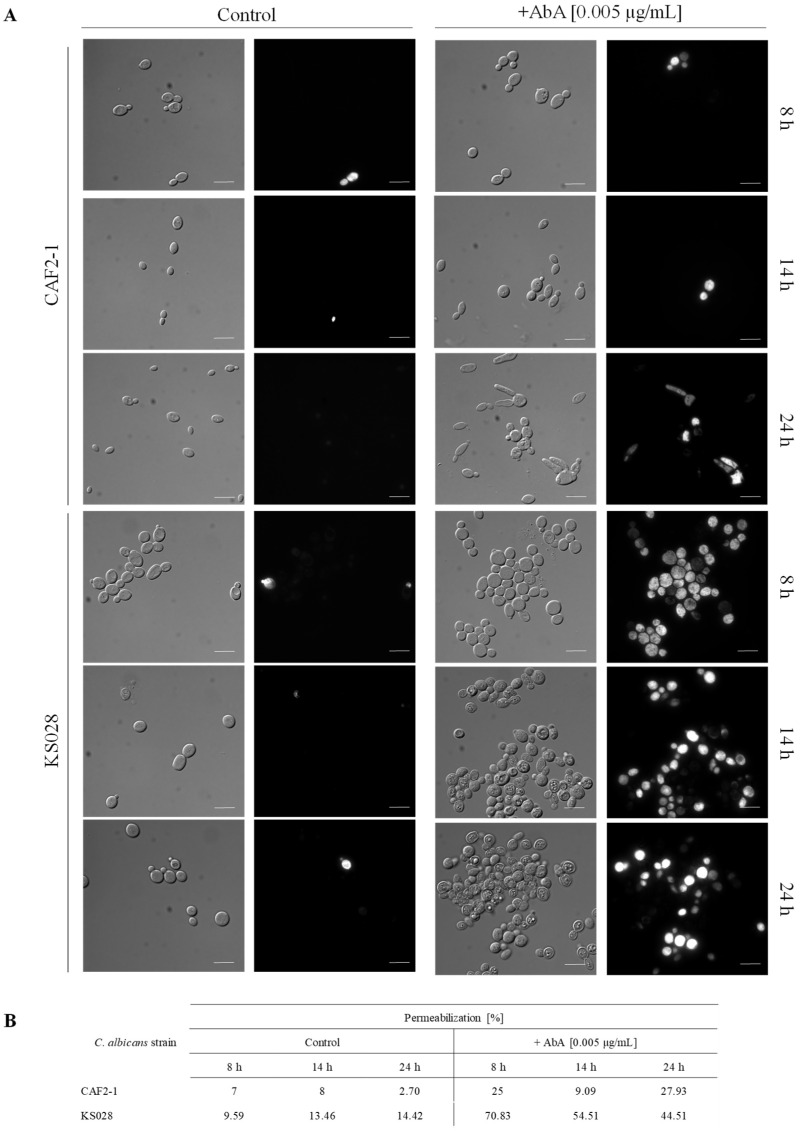
Permeabilization of *C. albicans* CAF2-1 and KS028 (*erg11∆/∆*) strains. (**A**) Representative micrographs of *C. albicans* CAF2-1 and KS028 staining with propidium iodide (PI) after 8, 14, and 24 h culturing without (control) or with AbA (0.005 µg/mL); scale bar: 10 μm. (**B**) Percent of permeabilized cells of *C. albicans* CAF2-1 and KS028 cultured with or without addition of AbA (0.005 µg/mL) (at least *n* = 100 cells per each case).

**Figure 7 ijms-23-09975-f007:**
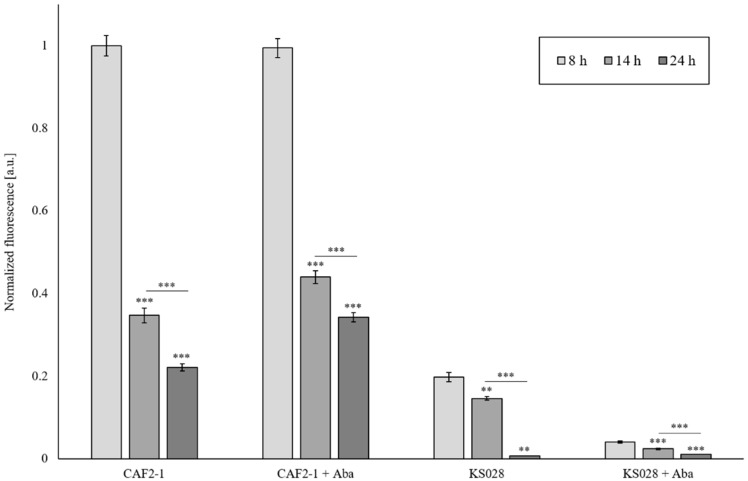
The efflux of rhodamine 6G (R6G) determined in 8, 14, or 24 h cultures, with or without the supplementation of 0.005 µg/mL Aureobasidin A (AbA) for *C. albicans* CAF2-1 and *C. albicans* KS028 (*erg11Δ/Δ*). Statistical analysis was performed by comparing rhodamine efflux between the *C. albicans* CAF2-1 and KS028 strains during different growth phases (** *p* < 0.01; *** *p* < 0.001).

**Table 1 ijms-23-09975-t001:** H^+^-ATPase activity measured by pH change after 200s., as determined for *C. albicans* CAF2-1 (WT) and KS028 (*erg11Δ/Δ*) strains. Cells for testing were taken after 8, 14, and 24 h of culture. The percentage of the difference between the number of units by which the pH decreased in the control (without the addition of AbA) and sample with the addition of AbA (0.005 µg/mL).

Strain/Time of Culture	8 h		14 h		24 h	
CAF2-1	control	+AbA	control	+AbA	control	+AbA
	1.5	1.81	2.02	1.81	2.44	1.45
pH range	7.495–5.995	7.495–5.685	7.495–5.475	7.495–5.405	7.495–5.055	7.495–6.045
pH change (%)	21%		−10.4%		−40.6%	
KS028	2.01	1.88	1.94	1.31	1.79	1.58
pH range	7.495–5.485	7.495–5.615	7.495–5.555	7.495–6.185	7.495–5.705	7.495–5.915
pH change (%)	−6.46%		−32.5%		−12%	

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
