# Peer review of "The Role of Ergosterol and Sphingolipids in the Localization and Activity of Candida albicans’ Multidrug Transporter Cdr1p and Plasma Membrane ATPase Pma1p"

_ijms, 2022, doi:10.3390/ijms23179975_

Round 1
Reviewer 1 Report
The authors of the manuscript titled "The role of ergosterol and sphingolipids in the localization and activity of Candida albicans multidrug transporter Cdr1p and plasma membrane ATpase Pma1p" report the role of ergosterol and sphingolipids in the localization and activity of Candida albicans multidrug transporter Cdr1p and plasma membrane ATPase Pma1p. The body of work presented here is appropriate for the International Journal of Molecular Sciences however, it needs some major revisions before it can be considered for publication.
Points need to be addressed
1. The authors should make the time points bigger in figure 1. For figure 1, at what growth point the diffusion time was measured, and does the values correlate between immunoblot and svFCS method?
2. For figure 2, if possible, the authors should provide a better-quality gel picture, and label the protein. Additionally, in the supporting information if the authors can provide SDS Page gels before putting them on the blot paper which provides information regarding the affinity of the antibodies for the proteins of interest.
3. For figure 4, can the authors speculate why the ergosterol levels go up from 8h to 14 h and then decrease at 24h. Additionally, why there is no increase in the levels of SLs was observed in the case of erg11 Δ/Δ as it is been mentioned in the discussion section of 2.3.
4. In the case of figure legend of figure 4, the SLs analysis is mentioned in ng/mg but the figures are in mg/mg. Please fix the mistake and increase the font size of the x and y axis for better visibility.
5. Can authors speculate why the reduced levels of sphingolipids are not necessary for delocalization of CaCdr1p and CaPma1p but are important when it comes to their activity? Is there another mechanism involved?
6. The conclusions are repeated twice. Please remove the conclusions in the results section. For section 5 conclusions please provide the details as a paragraph rather than bullet points.
7. In section 4, materials and methods, the authors need to provide information regarding whether the experiments were performed in duplicates or triplicates. The number of independent experiments should be given and provide information regarding whether these numbers are replicates within a single assay.
8. The authors need to provide a new figure S1. The new figure needs to be of better clarity and colors for visibility.
9. The supporting information section needs a title, author list, and table of content. The original images of the blots and gels need to be added to the supporting information with figure legends. These gels and blots need to be of better quality, label the proteins and the ladders need to be added.
Author Response
Reviewer 1:
The authors of the manuscript titled "The role of ergosterol and sphingolipids in the localization and activity of Candida albicans multidrug transporter Cdr1p and plasma membrane ATpase Pma1p" report the role of ergosterol and sphingolipids in the localization and activity of Candida albicans multidrug transporter Cdr1p and plasma membrane ATPase Pma1p. The body of work presented here is appropriate for the International Journal of Molecular Sciences however, it needs some major revisions before it can be considered for publication.
Points need to be addressed
- The authors should make the time points bigger in figure 1. For figure 1, at what growth point the diffusion time was measured, and does the values correlate between immunoblot and svFCS method?
Thank you for those suggestions. We inserted bigger time points in the Figure 1 and inserted values on the Y axis in a more readable scale. The diffusion time was analyzed after 24 hours of C. albicans CAF2-1 and erg11Δ/Δ cultures. This information was added in Figure 1 description. The results indicate a correlation between immunoblot and svFCS. We've added the right comment which we marked in blue.
- For figure 2, if possible, the authors should provide a better-quality gel picture, and label the protein. Additionally, in the supporting information if the authors can provide SDS Page gels before putting them on the blot paper which provides information regarding the affinity of the antibodies for the proteins of interest.
Thank you for your suggestions, the small amount of Cdr1 protein in the fractions may suggest that the gels are of poor quality, however, such results have already been published in the team of both ours (PMID: 31546699) and others (PMID: 14665469). We added the label of mass of proteins on the Figure 2.
Unfortunately we cannot provide stained SDS – PAGE gels, due to the fact, that most of the material were subjected to gels which were later proceeded along western blot protocol with anti-Cdr1 and anti-GFP antibodies. Nevertheless, we performed Ponceau S staining for selected experiments, (an example we shown on Fig. S2 which we added to supplementary materials as reviewer suggests), which shows the proteins presence in around 30% of Optiprep content. During preliminary tests we acknowledged that despite of using high amount of cells (around 100 OD) the protein content in fractions from DRMs isolation method is fairly low (what was expected, as we isolate only small fraction of overall cell proteins content). Because of Ponceau S staining result, together with later confirmation of high Pma1-GFP (which is known marker of DRMs) content in these fractions, we assumed that procedure of DRM isolation was efficient, and we were able to isolate DRM-associated proteins.
- For figure 4, can the authors speculate why the ergosterol levels go up from 8h to 14 h and then decrease at 24h. Additionally, why there is no increase in the levels of SLs was observed in the case of erg11 Δ/Δ as it is been mentioned in the discussion section of 2.3.
Thank you for asking this question. Previously we reported the similar effect of “bell-shaped” ergosterol content for C. albicans CAF2-1 WT strain in context of growth phases [PMID: 31546699]. So here we observed a typical increase in ergosterol level in late logarithmic phase of growth and then decrease in stationary phase of growth. In addition, Gutarowska and Żakowska observed a similar effect in other fungi [PMID: 19291213]. The corresponding comment has been added and marked in blue.
- In the case of figure legend of figure 4, the SLs analysis is mentioned in ng/mg but the figures are in mg/mg. Please fix the mistake and increase the font size of the x and y axis for better visibility.
Thank you for pointing out our mistake. We already changed the unit in Y axis in Figure 4B from „µg/mg” to „ng/mg”. We also increased font size of legend, X and Y axis in Figure 4A and 4B for better visibility according to your suggestion.
- Can authors speculate why the reduced levels of sphingolipids are not necessary for delocalization of CaCdr1p and CaPma1p but are important when it comes to their activity? Is there another mechanism involved?
Thank you for your question. We have tried to describe our line of thought and conclusions at the end of the Discussion chapter (lines: 379-389).
- The conclusions are repeated twice. Please remove the conclusions in the results section. For section 5 conclusions please provide the details as a paragraph rather than bullet points.
We have made every effort to ensure that the conclusions are not repeated in the text of the publication. In section 5 we changed the conclusions from bullet points for paragraph.
- In section 4, materials and methods, the authors need to provide information regarding whether the experiments were performed in duplicates or triplicates. The number of independent experiments should be given and provide information regarding whether these numbers are replicates within a single assay.
Thank you for this comment. We improved our Material and Methods section by adding an information about number of replicates in each experiment. Material and Methods section was supplemented with that information in following lines: 418-419 (section 4.3.), 435-436 (section 4.4.), 439 (section 4.5.), 461 (section 4.6.), 483 (section 4.7.), 506-507 (section 4.8.), 517-523 (section 4.10.), 550 (section 4.12) and 553-554 (section 4.13).
- The authors need to provide a new figure S1. The new figure needs to be of better clarity and colors for visibility.
Thank you for this remark. We improved our Figure S1 in order to provide better clarity and visibility of presented data. On the new Figure S1 we increased font size of legend, X and Y axis. We also colored bars in order to clarify presented data.
- The supporting information section needs a title, author list, and table of content. The original images of the blots and gels need to be added to the supporting information with figure legends. These gels and blots need to be of better quality, label the proteins and the ladders need to be added.
Accordingly to suggestions, we have formatted the suplementary materials. The original images of the blots and gels were added as Figure S2 and appropriately described in the text ms (marked in blue).
Reviewer 2 Report
In the manuscript entitled "The role of ergosterol and sphingolipids in the localization and 2 activity of Candida albicans’ multidrug transporter Cdr1p and 3 plasma membrane ATPase Pma1p”, the authors showed that CaCdr1p localizes in detergent resistant microdomains (DRMs) fraction in both C. albicans WT and erg11Δ/Δ strains. They further examined the sensitivity of the C. albicans strains to the inhibitor of SLs synthesis, Aureobasidin A (AbA). Interestingly, the C. albicans erg11Δ/Δ strain is more sensitive to the inhibitor of AbA that is caused by disruption of PM integrity in C. albicans erg11Δ/Δ but not in the C. albicans WT strain.
While the presented manuscript is covering an interesting topic, the authors need to perform further experiments (see below) and broaden the view for clinical use of the compound especially in the context of the bioavailability and pharmacokinetics and discuss reachable plasma levels required for the antifungal effect. I hope that the authors can provide a revised manuscript addressing all my concerns.
In fig. 2 the authors showed the viability of C. albicans CAF2-1 and erg11∆/∆ strains in the presence of AbA. While the authors calculated the MIC50 and MIC90 values, they do not show the raw data. Please include the dose-response curves that were used to calculate the MIC50 and MIC90 values and improve the figure quality (especially for the Growth phenotypes of C. albicans.
The authors further observed a change in the lipid composition of the PM in C. albicans via GC-MS method. Recently, a similar observations were made in mammalians (PMID: 32975484, PMID: 34919035). Here, the FIASMA fluoxetine seems to effect PM cholesterol levels by sequestering cholesterol in the endolysosomal compartment which seems to influence virus entry of SARS-CoV-2, influenza virus, and Ebola virus. Is a similar mechanism possible in fungi? The authors should inspect the subcellular localization of the lipids and perform co-localization assays and analyze them via Manders' Overlap Coefficient (MOC). Please use the mentioned PMIDs for the discussion. This might also explain the decrease in ergosterol levels upon AbA treatment.
In fig. 6, the authors showed images that examined the permeability of the fungi PM by using propidium iodide. Why do the authors observed less fungi growth in the untreated control of KS028 compared to AbA treatment? Please also add a quantification of the permeability images.
The authors also analyzed the H+-ATPase activity in C. albicans CAF2- 2911 (WT) and KS028 (erg11Δ/Δ) strains. Interestingly, they observed a decrease in acidification of the environment by cells treated with AbA. This is particular interesting as the FIASMA fluoxetine was also reported to influence pH values in the endolysosomal compartment of mammalians ((PMID: 32975484, PMID: 34919035). Does AbA treatment effects the stabilization of H+-ATPase on the cell surface? In figure 5, its seems that the plasma membrane ATPase Pma1p is re-localized upon 14 h of Aba treatment. In the respective table for the ATPase activity, does the authors show the pH values? If not please add them (the shown values are very low).
I have also some concerns about the statistical test used in the presented manuscript. The authors used through the entire manuscript unpaired Student’s t-test, however in some cases it is recommended to use Oneway-ANOVA with Tukey post-test. Please carefully reanalyze the data.
I hope that the authors can provide a revised manuscript addressing my concerns.
Author Response
Reviewer 2:
In the manuscript entitled "The role of ergosterol and sphingolipids in the localization and 2 activity of Candida albicans’ multidrug transporter Cdr1p and 3 plasma membrane ATPase Pma1p”, the authors showed that CaCdr1p localizes in detergent resistant microdomains (DRMs) fraction in both C. albicans WT and erg11Δ/Δ strains. They further examined the sensitivity of the C. albicans strains to the inhibitor of SLs synthesis, Aureobasidin A (AbA). Interestingly, the C. albicans erg11Δ/Δ strain is more sensitive to the inhibitor of AbA that is caused by disruption of PM integrity in C. albicans erg11Δ/Δ but not in the C. albicans WT strain.
While the presented manuscript is covering an interesting topic, the authors need to perform further experiments (see below) and broaden the view for clinical use of the compound especially in the context of the bioavailability and pharmacokinetics and discuss reachable plasma levels required for the antifungal effect. I hope that the authors can provide a revised manuscript addressing all my concerns.
In fig. 2 the authors showed the viability of C. albicans CAF2-1 and erg11∆/∆ strains in the presence of AbA. While the authors calculated the MIC50 and MIC90 values, they do not show the raw data. Please include the dose-response curves that were used to calculate the MIC50 and MIC90 values and improve the figure quality (especially for the Growth phenotypes of C. albicans.
Thank you for this remarks. According to your suggestions in Figure 3A we show the dose-response curves which were used for determination of MIC50 and MIC90 values. We also improved the quality of Figure 3B to make growth phenotypes of C. albicans more visible.
The authors further observed a change in the lipid composition of the PM in C. albicans via GC-MS method. Recently, a similar observations were made in mammalians (PMID: 32975484, PMID: 34919035). Here, the FIASMA fluoxetine seems to effect PM cholesterol levels by sequestering cholesterol in the endolysosomal compartment which seems to influence virus entry of SARS-CoV-2, influenza virus, and Ebola virus. Is a similar mechanism possible in fungi? The authors should inspect the subcellular localization of the lipids and perform co-localization assays and analyze them via Manders' Overlap Coefficient (MOC). Please use the mentioned PMIDs for the discussion. This might also explain the decrease in ergosterol levels upon AbA treatment.
Thank you for this question. It is very interesting that fluoxetine affects PM cholesterol level by sequestering cholesterol in the endolysosomal compartment. According to our results, we conclude that the possibility of presence of similar effect is small. This is supported by the results presented in Figure 4A – the ergosterol level (determined only for isolated PM) in C. albicans WT strain after treatment with AbA was comparable to control conditions (without addition of AbA) in all tested time points of culture (8, 14 and 24 hours). Thus, AbA do not affect ergosterol level in the PM of C. albicans in contrast to SLs level. If the mechanism would be similar to that presented in PMID: 32975484 (delocalization of cholesterol from PM to endosomes), the PM ergosterol level would decrease among time of the culture with addition of AbA.
Additionally, the fluoxetine is responsible for inhibition of acid sphingomyelinase (FIASMA) which consequences in decreased degradation of sphingomyelin in mammalian cells. In our study we used AbA which is an inhibitor of inositol phosphorylceramide (IPC) synthases and AbA treatment results in overall diminished SLs (IPC and M(IP)2C) level in fungal cells. Those compounds (fluoxetine and AbA) have a different mode of action on lipid metabolism which make them difficult to compare.
In fig. 6, the authors showed images that examined the permeability of the fungi PM by using propidium iodide. Why do the authors observed less fungi growth in the untreated control of KS028 compared to AbA treatment? Please also add a quantification of the permeability images.
Thank you for this questions. The Figure 6 constitutes the representative micrographs of C. albicans PI staining after 8, 14 or 24 hours of culture with or without addition of AbA. In order to prepare the microscope slides, after staining with PI C. albicans cells were harvested, dissolved in PBS and then concentrated for microscope observations. We observed a decreased growth in case of KS028 (erg11Δ/Δ) strain after AbA treatment. The impression of less growth in the untreated control of KS028 compared to AbA treatment is caused by the fact that treatment with AbA results in creation of cell clumps which is shown in Figure 6A. In order to clarify obtained results we provide the quantification of the percent (%) of permeabilized cells in all tested conditions.
The authors also analyzed the H+-ATPase activity in C. albicans CAF2- 2911 (WT) and KS028 (erg11Δ/Δ) strains. Interestingly, they observed a decrease in acidification of the environment by cells treated with AbA. This is particular interesting as the FIASMA fluoxetine was also reported to influence pH values in the endolysosomal compartment of mammalians ((PMID: 32975484, PMID: 34919035). Does AbA treatment effects the stabilization of H+-ATPase on the cell surface? In figure 5, its seems that the plasma membrane ATPase Pma1p is re-localized upon 14 h of Aba treatment. In the respective table for the ATPase activity, does the authors show the pH values? If not please add them (the shown values are very low).
Thank you for very interesting question. We interpret the results presented in figure 5A as a delay in the relocation of Pma1-GFP with PM rather than a re-location. In addition, the level of Pma1-GFP in PM after treatment for 14h AbA appears to be very similar or the same as for control culture without AbA. As we have shown in our previous work, Pma1p is more stable and seems to persists in PM longer than Cdr1p [PMID: 31546699]. In our previous work, western blot analyses with Pma1p and Cdr1p in isolated PM show that the Pma1p level is higher than Cdr1p [PMID: 31546699]. Additionally, as we write earlier, fluoxetine and AbA have a different mode of action on lipid metabolism which make them difficult to compare. As reviewer suggests, in Table 1, we added pH values.
I have also some concerns about the statistical test used in the presented manuscript. The authors used through the entire manuscript unpaired Student’s t-test, however in some cases it is recommended to use Oneway-ANOVA with Tukey post-test. Please carefully reanalyze the data.
Thank you for this suggestion. For the results presented on figures 4 and S1, we used One-way-ANOVA with Tukey post-test in place of unpaired Student's t-test.
I hope that the authors can provide a revised manuscript addressing my concerns.
Round 2
Reviewer 1 Report
The authors have addressed all the comments, and as a result, the quality of the manuscript was considerably improved.
Reviewer 2 Report
The authors have improved the manuscript accordingly to the suggestions.